# Extracting patient-level data from the electronic health record: Expanding opportunities for health system research

Erica Farrand[1]◉*, Harold R. Collard[1]◉, Michael Guarnieri[2]‡, George Minowada[2]‡, Lawrence Block[3]◉, Mei Lee[3]◉, Carlos Iribarren[3]◉

**1** Department of Medicine, University of California San Francisco, San Francisco, CA, United States of America, **2** Kaiser Permanente Medical Group, Kaiser Permanente Northern California, Oakland, CA, United States of America, **3** Division of Research, Kaiser Permanente Northern California, Oakland, CA, United States of America

◉ These authors contributed equally to this work.
‡ MG and GM also contributed equally to this work.
* erica.farrand@ucsf.edu

## Abstract

### Background

Epidemiological studies of interstitial lung disease (ILD) are limited by small numbers and tertiary care bias. Investigators have leveraged the widespread use of electronic health records (EHRs) to overcome these limitations, but struggle to extract patient-level, longitudinal clinical data needed to address many important research questions. We hypothesized that we could automate longitudinal ILD cohort development using the EHR of a large, community-based healthcare system.

### Study design and methods

We applied a previously validated algorithm to the EHR of a community-based healthcare system to identify ILD cases between 2012–2020. We then extracted disease-specific characteristics and outcomes using fully automated data-extraction algorithms and natural language processing of selected free-text.

### Results

We identified a community cohort of 5,399 ILD patients (prevalence = 118 per 100,000). Pulmonary function tests (71%) and serologies (54%) were commonly used in the diagnostic evaluation, whereas lung biopsy was rare (5%). IPF was the most common ILD diagnosis (n = 972, 18%). Prednisone was the most commonly prescribed medication (911, 17%). Nintedanib and pirfenidone were rarely prescribed (n = 305, 5%). ILD patients were high-utilizers of inpatient (40%/year hospitalized) and outpatient care (80%/year with pulmonary visit), with sustained utilization throughout the post-diagnosis study period.

### Discussion

We demonstrated the feasibility of robustly characterizing a variety of patient-level utilization and health services outcomes in a community-based EHR cohort. This represents a

**Data Availability Statement:** Data Availability: The dataset used in this study is held by the Kaiser Permanente Northern California Division of Research. Any researcher interested in accessing

this dataset can submit an application form through the Research Collaboration Portal (rcp. kaiserpermanente.org) requesting access. Please contact the research collaboration portal staff (Email: Victoria.k.peckham@kp.org) for further assistance. All relevant data are within the paper.

**Funding:** This study was funded by an Investigator Initiated Study grant from Genentech, HL12131 awarded to HRC and CI. https://www.gene.com/ The funders had no role in study design, data collection and analysis, decision to publish, or preparation of the manuscript. The Pulmonary Fibrosis Foundation awarded a research grant to EF. https://www.pulmonaryfibrosis.org/ researchers-healthcare-providers/research-opportunities/scholars. The funders had no role in study design, data collection and analysis, decision to publish, or preparation of the manuscript.

**Competing interests:** We have read the journal's policy and one of the authors of this manuscript (Dr. Harold Collard) has the following competing interests: personal fees from Advanced Medical, Polerean, and WebMD, and grants from Pulmonary Fibrosis Foundation outside of the submitted work. These entities provided no funding and had no role in the design, preparation, or submission of this manuscript. The competing interest does not alter our adherence to PLOS ONE policies on sharing data and materials.

substantial methodological improvement by alleviating traditional constraints on the accuracy and clinical resolution of such ILD cohorts; we believe this approach will make community-based ILD research more efficient, effective, and scalable.

## Introduction

Interstitial lung diseases (ILDs) are a diverse group of diffuse parenchymal lung disorders that affect approximately 250,000 people in the United States and result in poor health-related quality of life, increased health care resource utilization, premature respiratory failure, and death [1–4]. Following U.S. Food and Drug Administration (FDA) approval of pharmacologic therapies for major ILD subtypes, increased attention has focused on evaluating treatment efficacy in real world settings [5–9]. However traditional approaches to community-based population health studies require a significant investment in personnel and infrastructure to support participant recruitment, enrollment, data collection and management, particularly in rare diseases such as ILD [10, 11]. As a result ILD research is primarily conducted in tertiary care populations, and there is a limited understanding of ILD diagnostics, management, and outcomes in community-based settings where the majority of patients access care. An innovative approach is required to make ILD research feasible in representative clinical practice settings.

The electronic health record (EHR) provides an efficient, effective, and scalable approach to real-world longitudinal cohort development. EHRs capture an unparalleled breadth and depth of clinical, quality, process, and outcome measures [12–15]. However secondary EHR data use relies heavily on unstructured data and manual extraction which limit its practical use [16, 17]. Automated structured and unstructured data capture is possible using EHR-based tools and algorithms [18]. The data can be rigorously monitored and validated, and the tools iteratively refined. EHR data have been successfully applied in other disease contexts [19–21]. Regional EHR data, with detailed patient-level information, has been particularly impactful in advancing delivery science in other contexts and stands to fundamentally improve population research in ILD [22–24].

In this study, we test whether fully automated data-extraction algorithms and natural language processing (NLP) can be applied to robustly characterize the diagnosis and management of ILD patients. We also highlight several important observations from this approach that we believe will catalyze ILD health research by supporting studies on ILD incidence, prevalence, and health services delivery beyond academic and specialty care centers.

## Materials and methods

The study population was derived from the Kaiser Permanente Northern California (KPNC) population, a non-profit, community-based, integrated health care delivery organization which includes 21 medical centers, 60 outpatient facilities, 110 outpatient pharmacies, and a centralized laboratory. KPNC is a regional healthcare system, currently providing care to over 4.5 million members, representing 30% of the population in the 14-county area of Northern California. Eligible patients were adult (age 18 and older) KPNC members receiving care between January 2012 and December 2019, ensuring all patients had the potential for at least one full year of follow-up. The chosen time frame also aligned with KPNC adoption of the current EHR system (EPIC Systems, Verona, WI). Institutional review boards at the University of California San Francisco (#14–15459), and the KPNC Division of Research (#CN-15-2126-H_05) approved the study protocol. The primary dataset was deidentified prior to access.

A subset of patient records was identified for algorithm validation. The IRB waived the requirement for informed consent for this retrospective study of medical records as many of the participants were deceased and the study was determined to pose minimal risk.

## Case identification

Patients with ILD were identified using a previously developed algorithm based on International Classification of Diseases (ICD) codes, ninth and tenth revisions [25]. This highly-specific algorithm requires cases to have at least two claims with an ILD code at least one month apart and chest computed tomography (CT) procedure code (ICD-9-CM 87.41 & CPT-4 71250, 71260, 71270) on or before the date of the second ILD code (S1 Table). Identified cases were censored at the time of death or loss to follow-up, the latter defined as 6 months or greater without an EHR encounter of any type. A random validation sample of 200 cases underwent a structured medical record review by an expert ILD clinician (E.F.) to confirm ILD diagnosis and ILD subtype in order to assess performance of the ILD algorithm.

## EHR data extraction

Data from the EHR were extracted and transformed into a common format according to the Virtual Data Warehouse and Observational Medical Outcomes Partnership (OMOP) Common Data Model [26]. Thereby allowing use of standardized analytics. Data describing baseline demographics and practice patterns at the time of diagnosis (e.g., use of limited autoimmune serologies, chest high-resolution computed tomography (HRCT), pulmonary function tests (PFT) and pathology) were extracted from structured data fields. Autoimmune serologies were limited to antinuclear antibodies, rheumatoid factor, and anti-cyclic citrullinated peptide, three tests recommended as part of a general serologic evaluation in patients with suspected interstitial lung disease that could be reliably extracted from the her [27]. Baseline PFT values included forced expiratory volume (FEV1), forced vital capacity (FVC) and the diffusing capacity (DLCO). Raw values were extracted directly from EHR respiratory flowsheets and percent predicted values were calculated for each individual patient using the Global Lung Function Initiative Network reference values to ensure standardization [28]. Pharmacy data were queried to determine which medications were used in the initial management of ILD. Corticosteroid prescriptions were limited to those reflecting long term use, defined as prescriptions for at least 30 consecutive days and a dose $\geq$ 20mg daily. Use of nintedanib and pirfenidone were only analyzed beginning in October 2014, corresponding to FDA approval of these medications. Utilization and outcomes data were extracted including follow up chest CTs and PFTs, outpatient visits, hospital admissions, and death records. All data were extracted from existing structured data fields or combination of fields when variable capture was redundant. Internal validation was used to assess variable completeness and concordance between extracted data and free text.

In order to extract data on the presence or absence of radiology "usual interstitial pneumonia" (UIP) pattern (a data element not captured in structured fields), unstructured data sources (e.g., free text) were searched using an open-source NLP system, Apache clinical Text Analysis and Knowledge Extraction System (cTAKES). Radiology reports were searched using the terms "usual interstitial pneumonia" and "UIP". The search algorithm reviewed the text immediately before and after the terms. If the terms were identified and no evidence of negation terms (e.g., "no" or "inconsistent with") was found, the CT was considered to demonstrate UIP pattern. If the terms were not found, or if the terms were found but evidence of negation terms was also found, the CT was considered to not demonstrate a UIP pattern. The NLP algorithm used regular expression and generalized Levenshtein edit distance to identify close

misspellings of the key terms of interest [29]. A subset of NLP results (40%) was manually reviewed by an ILD expert (E.F.) and validated in order to assess the performance of the NLP algorithm.

## Statistical analysis

Data processing and analysis were performed in R (version 1.4.1717). The positive predictive value (PPV) and binomial 95% confidence interval (CI) were determined for the ILD algorithm, ILD diagnosis algorithm, and CT radiology NLP algorithm. The odds ratio (OR) of detecting a UIP pattern on a HRCT radiology report versus conventional CT was calculated using logistic regression. For longitudinal health care resource utilization outcomes, we analyzed each year of utilization separately to test whether the results varied over time. We calculated per-comparison P values and pre-specified family-wise adjusted p values to account for multiple comparisons. P <0.05 was considered significant, and all P values were 2 sided.

## Results

We identified a total of 5,399 KPNC members (a prevalence of 118 per 100,000) between January 2012 and December 2019 with ILD (referred to subsequently as the ILD Cohort), distributed widely throughout the 14-county area of Northern California (S1 Fig, S2 Table). Twenty-five percent (n = 1,350) of cases underwent HRCT as part of their diagnostic evaluation, with the remaining 75% receiving a conventional CT chest. Pulmonary function testing (71%, n = 3821) and limited autoimmune serologies (54%, n = 2896) were commonly performed during the diagnostic evaluation (Table 2). Only 12% (n = 642) of patients had a lung biopsy of any type; of these 43% (n = 276) were surgical biopsies and 57% (n = 366) were bronchoscopic biopsies (not further characterized). Nineteen percent (n = 1004) of patients had a bronchoscopy that was not associated with a biopsy.

The ILD Cohort was 54% female, 79% 60 years of age or older, and 59% white (Table 1). The majority of patients were former smokers (51%) and lived in urban (49%) or suburban (26%) locations. The mean (± Standard Deviation (SD)) FVC percent of predicted was 75.19 ±18.92%, the mean FEV1 percent of predicted was 73.70±20.12%, and the mean DLCO percent of predicted was 51.40 ± 16.70% (Table 2).

Structured case validation revealed a PPV of the ILD algorithm of 95.5% (95% confidence interval (CI), 95.38, 95.61). The PPV for identifying a specific diagnosis among the cases of ILD was 74% (95% CI, 73.93, 75.53). The most common ILD diagnosis identified was idiopathic pulmonary fibrosis (18%), followed by connective tissue disease related ILD (12%) and chronic hypersensitivity pneumonitis (10%). One-third of the ILD Cohort (33%) had ILD that did not list a specific diagnosis (S2 Fig). Median survival estimate of the full ILD cohort was 72 months (S3 Fig).

Overall, 99% (n = 5366) patients in the ILD Cohort had CT reports that could be reviewed using NLP, of which 11% (n = 590) were classified as UIP pattern and 89% (n = 4776) were classified as not UIP (Fig 1A). Of the 5366 CT reports, 40% were manually reviewed (n = 2168) of which 14% (n = 312) had a UIP pattern and 86% (n = 1856) did not have a UIP pattern. This corresponds to a PPV for detecting a UIP pattern of 94.29% (90.70, 96.48%). During validation of the NLP algorithm it was observed that of the CTs classified as not UIP (86%, n = 4776), 49% (n = 1053) specified no UIP pattern while a distinct pattern was not specified for the remaining 37% (n = 803). Detection of radiologic patterns differed significantly between reports from HRCT and conventional CTs (Fig 1B). The OR of a UIP pattern in the HRCT group (n = 501) versus the conventional CT scan group (n = 1667) was 4.83 (95% CI 3.76, 6.21). On the other hand, patients who received a HRCT were five times less likely to

**Table 1. Characteristics of patients identified by the ILD-algorithm.**

| Patient Characteristics | Total N = 5,399 |
|---|---|
| Sex at birth | |
| Female | 2,914 (54%) |
| Age, yrs | |
| 18–29 | 40 (1%) |
| 30–39 | 99 (2%) |
| 40–49 | 266 (5%) |
| 50–59 | 701 (13%) |
| 60–69 | 1350 (25%) |
| $\geq 70$ | 2941 (54%) |
| Race | |
| Asian | 680 (13%) |
| Black | 399 (7%) |
| Multi-racial | 341 (6%) |
| Native American | 31 (1%) |
| Unknown | 733 (14%) |
| White | 3,188 (59%) |
| Ethnicity | |
| Hispanic | 926 (21%) |
| Non-Hispanic | 3494 (79%) |
| Smoking History | |
| Current | 175 (3%) |
| Former | 2763 (51%) |
| Never | 2461 (46%) |
| Supplemental Oxygen (at time of dx) | 665 (12%) |
| Geographic Location | |
| Urban | 2,655 (49%) |
| Suburban | 1,1416 (26%) |
| Metro | 992 (18%) |
| Rural | 20 (0%) |

Values are No. (%)

have an indeterminate pattern on report than those who received a conventional CT (OR = 0.21, 95% CI (0.16, 0.27)).

Long-term corticosteroid use was the most common medication used in the management of ILD patients following diagnosis (17%, n = 911), followed by mycophenolate mofetil (13%, n = 680). Only two percent (n = 131) was prescribed nintedanib and 3% (n = 174) was prescribed pirfenidone (Table 3).

Health care utilization was relatively constant among surviving patients throughout the post-diagnosis study period for patients with ILD (Fig 2). There were no significant differences in percentage of at risk ILD patients utilizing healthcare for all outcomes evaluated (all P > 0.05). On average, approximately 4 in 5 ILD patients saw a pulmonologist at least once a year, approximately half of ILD patients visited the emergency department at least once a year, and approximately 2 in 5 ILD patients were hospitalized at least once a year. The majority of ILD patients underwent regular PFT testing as part of their longitudinal care while the minority underwent regular chest CTs. The number of patients at risk over time decreased substantially due to a combination of lost-to-follow up, right censoring and death.

**Table 2. ILD diagnostic evaluation and study results.**

| Diagnostic Studies and Results | Total N = 5399 |
| --- | --- |
| Autoimmune Serologies | 2,896 (54%) |
| Antinuclear Antibody | 2220 (41%) |
| Rheumatoid Factor | 2483 (46%) |
| Anti-cyclic Citrullinated Peptide | 1715 (31%) |
| CT Chest | 5,399 (100%) |
| HRCT | 1,350 (25%) |
| UIP-like fibrotic pattern on CT Chest* | 594 (11%) |
| Pulmonary Function Test | 3821 (71%) |
| FEV1, mean value | 1.97 ± 0.63 |
| FEV1, percent of predicted value | 75.19 ± 18.92 |
| FVC, mean value (ml) | 2.47 ± 0.86 |
| FVC, percent of predicted value | 73.70 ± 20.12 |
| DLCO, mean value (mmol/min/kPa) | 11.53 ± 4.50 |
| DLCO, percent of predicted value | 51.40 ± 16.70 |
| 6 Minute Walk Test | 1093 (20%) |
| Lung Biopsy | 642 (12%) |
| Surgical Biopsy | 276 (5%) |
| Bronchoscopy | 1004 (19%) |

Values are mean ± SD of No. (%). *N = 5366. HRCT = high-resolution computed tomography; UIP = usual interstitial pneumonia; FEV1 = forced expiratory volume; FVC = forced vital capacity; DLCO = diffusing capacity of the lung for carbon monoxide.

## Discussion

Ensuring high-quality evidence-based care in ILD requires defining and characterizing disease epidemiology, healthcare utilization, and practice patterns in real world settings. Such efforts have historically relied on large-scale recruitment efforts and manual data collection methods that are separate from the clinical enterprise and present a substantial barrier to success. In our study, we aimed to bypass this barrier by applying accurate, automated, and scalable data extraction methodology to a community-based, real world EHR. Our results demonstrate that a code-based EHR algorithm can be used to accurately identify a cohort of ILD patients. ILD subtypes have previously been identified using algorithms, however this is the first study to target a broader cohort of ILD patients more relevant to clinical practice. We also describe the process of building a robust longitudinal ILD patient cohort using baseline, process, and outcome data available in the EHR. We included variables commonly collected in patient registries and clinical trials, as well as data reflecting healthcare utilization and practice patterns. This expanded variable list can be reliably and automatically extracted from the EHR. Further, our results demonstrate that unstructured data sources can be automatically processed through the application of NLP to chest CT reports.

More broadly, our results demonstrate the unique power of the EHR to transform health research. Unlike traditional tertiary cohorts and voluntary registries, community healthcare system EHR-based studies ground our study of ILD diagnosis and management in the real world. Further, EHR-based studies directly inform and enable subsequent implementation efforts to establish best practices in clinical care. Once an automated EHR-based cohort is developed, it can be easily reanalyzed at intervals to assess the impact of clinical interventions on practice patterns and patient outcomes. This pairing of research with care improvement

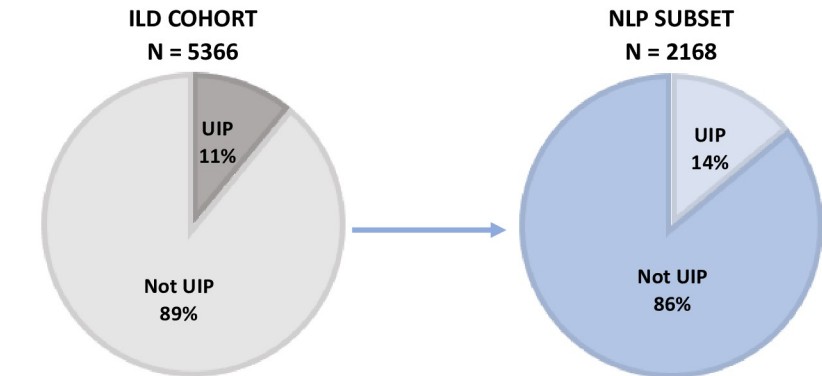

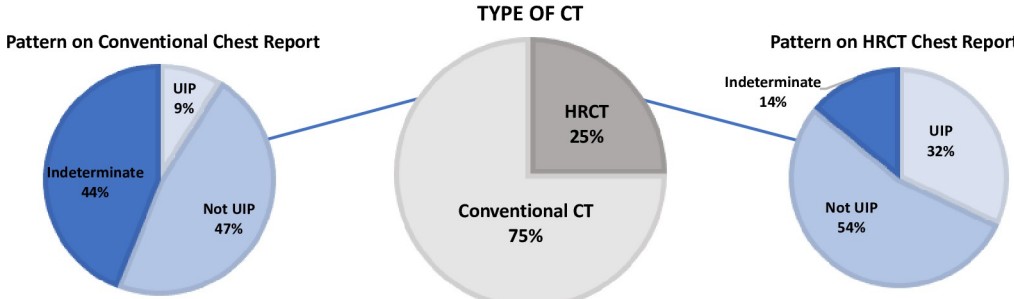

**Fig 1. a**. Pattern Detection on CT Chest Reports Using Natural Language Processing. Radiographic pattern detection on CT Chest reports from ILD Cohort: detection of usual interstitial pneumonia (UIP) and not UIP patterns using natural language processing (NLP) and manual validation. **b**. Comparison of Radiographic Patterns in HRCT vs. Conventional CT. Radiographic pattern detection on CT Chest reports from ILD Cohort: comparison of pattern detection in high-resolution CT (HRCT) reports vs. conventional CT reports.

through the EHR is at the heart of what the National Academies has called the Learning Healthcare System [30], and it holds great promise for quality improvement and public health for patients with ILD.

We believe the data reported in this study demonstrate that ILD care in the KPNC system is of high quality. The epidemiology, diagnostic evaluation, and management utilization mirrors a number of the findings from tertiary expert centers [3, 4, 31–33]. As importantly however, these data suggest several areas for care improvement. We highlight a few examples below.

First, an ILD diagnosis is not achieved in a sizable subgroup of patients with ILD. We hypothesize that this may in part stem from underutilization of guideline-recommended diagnostic studies, in particular HRCT. While one hundred percent of the cohort had a CT Chest performed on or before the time of diagnosis (this was part of our case definition), only one quarter had a HRCT as recommended by ILD guidelines. We observed significant differences in the detection of UIP pattern in HRCT reports as compared to conventional CT. We hypothesize that this finding is impacted by both differences in pretest probability of ILD in patients receiving an HRCT vs. conventional CT, as well as differences in test characteristics (i.e. CT precision, experience of radiologists). Overall, resolving knowledge gaps and operational barriers to the use of key diagnostic strategies, such as HRCT, specialty referral, and

**Table 3. ILD management and monitoring.**

| Medications | Total N = 5399 |
|---|---|
| Corticosteroids | 911 (17%) |
| Mycophenolate | 680 (13%) |
| Azathioprine | 239 (4%) |
| Rituximab | 169 (3%) |
| Pirfenidone* | 174 (3%) |
| Nintedanib* | 131 (2%) |
| Cyclophosphamide | 15 (0%) |
| Cyclosporine | 18 (0%) |
| Monitoring | Total N = 5399 |
| Pulmonary Visits | 4,446 (82%) |
| CT Chest | 3600 (67%) |
| Pulmonary Function Test | 3,119 (57%) |
| Supplemental Oxygen | 1,775 (32%) |
| Echocardiogram | 1,767 (32%) |
| Pulmonary Rehabilitation | 190 (3%) |

Values are No. (%)

* Utilization analyzed beginning in October 2014, corresponding to FDA approval of these medications.

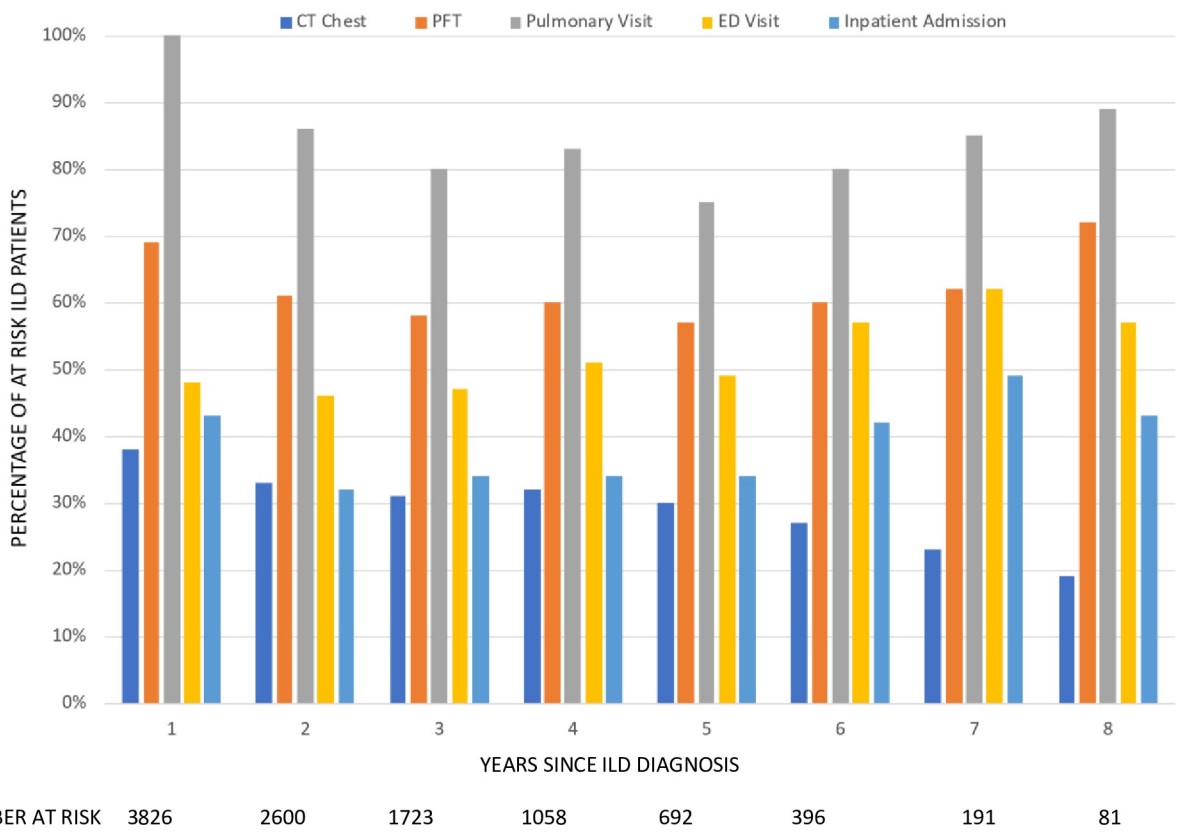

**Fig 2. Longitudinal health care resource utilization in ILD.** Annual percentages of health care utilization for patients with ILD post-diagnosis. PFT = pulmonary function test; ED = emergency department.

multidisciplinary case conference discussions, in community-based settings may be highly impactful in improving ILD diagnostic precision.

Second, the use of long-term corticosteroids in patients with ILD was 17%, while the use of the nintedanib and pirfenidone was 5% combined. While we expect occasional short-term corticosteroid use in an ILD cohort, their long-term use is associated with substantial morbidity and better tolerated, safer treatment options (e.g., mycophenolate mofetil) exist [34, 35]. Understanding what is driving long-term corticosteroid use and defining when and how non-steroidal immunomodulatory agents such as mycophenolate mofetil are prescribed will help to improve alignment with best practice. Nintedanib and pirfenidone are recommended first-line therapy for patients with idiopathic pulmonary fibrosis and are known to be effective in other forms of progressive fibrosing ILD [36–38]. Understanding patient, provider, and system-level barriers to the use of nintedanib and pirfenidone is a necessary first step to expanding guideline-based use of these medications and developing targeted clinical decision-making support.

Third, high rates of health care utilization are sustained at least eight years post-diagnosis in patients with ILD. This finding expands on our initial study limited to IPF patients, in which we demonstrated significantly higher rates of utilization in IPF patients compared to controls for five years post diagnosis [25]. Sustained health care utilization throughout the disease course suggests the need to develop and implement longitudinal care models and decision aids that meet the shared complex needs of ILD patients. These include guidance on process of care, medication choice and adjustment, management plans for acute worsening of symptoms, longitudinal evaluation of pulmonary function and HRCT, and end of life care planning.

There are limitations to this study. First, in order to develop methods for extracting ILD data from the EHR, we analyzed a single integrated health care system. At this stage algorithms must be developed and/or tailored for a specific health system's data. However, the algorithms and tools applied in this study were intentionally designed to pull from common EHR data elements, use common free-text terminology, and apply standardized data processing and analytic tools (e.g. OMOP and cTAKES) in order to facilitate the future development of similar EHR-based registries in other health systems. Second, our algorithm validation process was based on retrospective case review. Although these cases were randomly selected for review, if the cases in the validation samples were systematically different than the remaining sample, we could have over or underestimated the PPVs of the algorithms. Third, limitations in our data set precluded analysis of other worthwhile topics including the types of physicians providing care (generalist vs specialist), the impact of multidisciplinary case conference discussion on the diagnosis and management of ILD, important utilization metrics including pulmonary rehabilitation and palliative care, and the extent to which patients accessed ILD care outside of the KPNC system. Future studies combining our real-world EHR cohort with other data sources can expand the types of health care delivery questions that can be answered.

## Conclusion

In summary, these results demonstrate the transformative value of an EHR-based ILD cohort derived from large, community-based practice. By applying automated data extraction tools to alleviate logistical and methodological constraints, such real-world data sets facilitate health research, catalyzing our ability define patient care patterns, evaluate variability in outcomes, identify evidence-practice gaps, and implement solutions.

## Supporting information

**S1 Table. ILD diagnostic codes and procedure codes.**
(DOCX)

**S2 Table. Primary KPNC facility.** Percentages of the ILD Cohort receiving care in each of the KPNC regions. Values are No. (%).
(DOCX)

**S1 Fig. KPNC 14 county area with regional facilities.** 14-county area of Northern California with regional facilities that make up the KPNC membership.
(TIF)

**S2 Fig. ILD diagnoses identified by the ILD algorithm.** BO = bronchiolitis obliterans; HP = hypersensitivity pneumonitis; IPF = idiopathic pulmonary fibrosis; IPAF = interstitial pneumonia with autoimmune features; ILD = interstitial lung disease; NSIP = nonspecific interstitial pneumonia; OP = organizing pneumonia.
(TIF)

**S3 Fig. Kaplan-Meier survival estimate.** Survival estimate of the full ILD cohort. The number of patients at risk over time decreased due to a combination of lost-to-follow up, right censoring and death.
(TIF)

## Acknowledgments

EF had access to all of the data in the study and takes responsibility for the integrity of the data and the accuracy of the data analysis. EF, HRC, and CI contributed substantially to the study design, data analysis and interpretation, and the writing of the manuscript. MG and GW contributed to data interpretation and manuscript preparation. LB and ML contributed to data collection and analysis. The authors would like to acknowledge Dr. Thomas Urbania who contributed to the development of the NLP algorithm, and Drs. Liping Liu and Kirtee Raparia who contributed to the collection of pathology data.

## Author Contributions

**Conceptualization:** Erica Farrand, Harold R. Collard, Carlos Iribarren.

**Data curation:** Erica Farrand, Lawrence Block, Mei Lee.

**Formal analysis:** Erica Farrand, Lawrence Block.

**Funding acquisition:** Harold R. Collard, Carlos Iribarren.

**Investigation:** Erica Farrand, Harold R. Collard.

**Methodology:** Erica Farrand.

**Project administration:** Mei Lee.

**Resources:** Michael Guarnieri, George Minowada, Mei Lee, Carlos Iribarren.

**Supervision:** Carlos Iribarren.

**Validation:** Erica Farrand.

**Writing – original draft:** Erica Farrand.

**Writing – review & editing:** Erica Farrand, Harold R. Collard, Michael Guarnieri, George Minowada, Carlos Iribarren.

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
