## [Decision Letter · Decision Letter 0]

2 Aug 2022

PONE-D-22-15208Extracting Patient-Level Data from the Electronic Health Record: Expanding Opportunities for Health System ResearchPLOS ONE

Dear Dr. Farrand,

Thank you for submitting your manuscript to PLOS ONE. After careful consideration, we feel that it has merit but does not fully meet PLOS ONE’s publication criteria as it currently stands. Therefore, we invite you to submit a revised version of the manuscript that addresses the points raised during the review process.

We look forward to receiving your revised manuscript.

Kind regards,

Nguyen Quoc Khanh Le

Academic Editor

PLOS ONE

Journal Requirements:

“Drs. Erica Farrand, Michael Guarnieri, George Minowada, Mr. Lawrence Block, and Ms. Mei Lee have declared that no competing interests exist. Dr. Harold Collard has read the journal's policy and has the following competing interests: personal fees from Advanced Medical, Polarean and WebMD, and grants from Pulmonary Fibrosis Foundation outside of the submitted work.”

Reviewers' comments:

Reviewer's Responses to Questions

**Comments to the Author**

1. Is the manuscript technically sound, and do the data support the conclusions?

Reviewer #1: Yes

Reviewer #2: Yes

Reviewer #3: Yes

2. Has the statistical analysis been performed appropriately and rigorously? 

Reviewer #1: Yes

Reviewer #2: I Don't Know

Reviewer #3: Yes

3. Have the authors made all data underlying the findings in their manuscript fully available?

Reviewer #1: Yes

Reviewer #2: No

Reviewer #3: Yes

4. Is the manuscript presented in an intelligible fashion and written in standard English?

Reviewer #1: Yes

Reviewer #2: Yes

Reviewer #3: Yes

5. Review Comments to the Author

Reviewer #1: The use of several data sources is a real strength, as is the wide range of data elements. The paper is well written.

Q. While they used NLP on the radiology reports, it was unclear to me if they used NLP on the EHR physician notes. Please clarify that. If not, why not? (This is not intended to be a barrier to publication, but we should know this.)

Note: The word data is a plural word. They write "the pharmacy data was queried..." No. it should be: "the pharmacy data were queried..."

In general, the authors have done a very good job presenting the findings, the methods, and the wide range of sources and data elements.

Reviewer #2: I enjoyed reading your paper. In it, you lay out an analysis of how it is possible to use a regional EHR to look at a disease state (ILD) by automating longitudinal ILD cohort development. The importance of the study is in the proof of principle in characterizing such a cohort for the sake of future studies of intervention. You show this well. Much of what I have to say critically is more about the fine points of how the characterization was successful or perhaps could be improved. Thank you for submitting this important work.

1. First, has a similar demonstration ever been done with regional EHR for a different disease, even outside of pulmonary medicine (cancer, cardiology, or rheumatology, for example)? Your discussion should comment on this and put this paper in context with what has already been done (or not done) in this area.

2. There was a very low number of patients with “IPF” who were reported to take antifibrotic medications. Please discuss. Was this due to lack of access to (or use of) ILD specialty centers?

3. The information included in “autoimmune serology” is limited. Guideline directed evaluation would include some additional tests (SSA, SSB, Jo-1 to name a few). What percentage had a full cadre of autoimmune serologies completed? What percent of the “IPF” patients had this evaluation?

4. Are there any specialty ILD programs in the KPNC system in this study? If so, how often were they engaged and were there differences in any of your findings based upon whether or not they had at least one visit with an ILD program? If not, is there any way to know from your data or from other past publications how often KPNC patients are seen at a regional ILD center?

5. On a similar note, it is quite surprising to me that 20% of patients with ILD did not see a pulmonologist (you state in the text that it was 4/5 that saw a pulmonologist – although Figure 2 shows that 100% of them saw a pulmonologist in year 1). Please clarify and discuss. Is there any clue that perhaps they saw a pulmonologist outside of the KPNC system? Was the significance of ILD not appreciated by the PCP or was the finding perhaps not even taken not of (I see this sometimes when a patient gets the CT for another reason such as a chest pain, nodule or aorta evaluation that the ILD never gets evaluated and the ordering physician just focuses on the initial indication for the scan). Could a sample of them have a chart review to learn if there was some explanation for this?

6. What were the results of the subset of patients that were chart reviewed by an ILD specialist? I can't find this information, sorry if I am missing it. I see the information about the CT review, but thought there was also intended to be a review of the clinical cases by chart review.

7. You state that “the epidemiology, diagnostic evaluation, and management utilization mirrors many of the findings from tertiary expert center”. I challenge that statement some, given differences in the use of HRCT, numbers of autoimmune serology tests, and frequency of PFT or spirometry use. And you have no data about it, but I strongly doubt that there was frequent utilization of multidisciplinary discussion (MDD) groups, which is central to the diagnostic evaluation at a tertiary ILD center. Please speak to this in your discussion and soften your statement if you agree. Also, the management seems to differ greatly from what would be done at a tertiary expert center.

8. The number of patients with IPF who did not have a UIP pattern on CT was quite high. This begs me to wonder whether a whole lot of these patients really had some other ILD but were declared to have IPF but were wrongly diagnosed. Please comment. A “baseline characteristics” table in the supplement with just “IPF” patients might shed some light on this, as many recent IPF studies have shown very consistent characteristics and so we can see how your study’s “IPF patients” are similar or different from other cohorts.

9. Was there a high correlation between the CT reports that were manually reviewed and determined to have UIP pattern and the 11% that would have been declared as UIP through NLP? If not fully correlated, then that may have some impact upon the operating characteristics you describe for UIP by NLP.

10. Was there a high correlation between the CT reports that were manually reviewed and determined to have UIP pattern and the 11% that would have been declared as UIP through NLP? If not fully correlated, then that may have some impact upon the operating characteristics you describe for UIP by NLP.

11. I find in my own practice that many radiology reports from a chest CT do not comment one way or another about presence of UIP. Did you consider having a sample of some of those (the ones with no comment one way or another about UIP) independently reviewed?

12. Were the PFT values available as discrete values in the EHR extraction, or were they manually extracted? I know that with my own EPIC EHR system, PFT values are not usually available as discrete values/fields, although that may well be different in another hospital system. Were spirometry values (ones that were not part of a full PFT) also available, as such?

13. Were spirometry values (ones that were not part of a full PFT) also available, as such?

Reviewer #3: It is an exciting study and promising approach. Few minor comments for discussion:

1. page 8, line 193: the Authors state that "11% ... were classified as UIP patterns (correlating with an IPF diagnosis)...". As UIP radiological pattern is not always unequivocally bound to IPF diagnosis, I would suggest omitting to refer to the clinical term of IPF;

2. Page 9, lines 206-209: it would be interesting to learn the prescription pattern in UIP vs non-UIP;

3. Table 1: surprisingly, rural geographic location was close to 0, and the Authors estimated the frequency of chronic HP to be 10% (the third most common diagnosis). This discrepancy may be worth commenting on;

4. Table 3: please consider ordering medications and monitoring tests according to frequency.

6. PLOS authors have the option to publish the peer review history of their article (what does this mean?). If published, this will include your full peer review and any attached files.

Reviewer #1: **Yes: **Ross Koppel

Reviewer #2: No

Reviewer #3: **Yes: **Wojciech J.Piotrowski

---

## [Author Response · Author response to Decision Letter 0]

20 Sep 2022

JOURNAL REQUIREMENT 1: Please provide additional details regarding participant consent. In the ethics statement in the Methods and online submission information, please ensure that you have specified (1) whether consent was informed and (2) what type you obtained (for instance, written or verbal, and if verbal, how it was documented and witnessed). If your study included minors, state whether you obtained consent from parents or guardians. If the need for consent was waived by the ethics committee, please include this information. 

RESPONSE TO REQUIREMENT 1: Our full ethics statement was previously included in the online submission form but not in the methods section. The statement has been added to the methods section and now reads:

Institutional review boards at the University of California San Francisco (#14-15459), and the KPNC Division of Research (#CN-15-2126-H_05) approved the study protocol. The primary dataset was deidentified prior to access. A subset of patient records was identified for algorithm validation. The IRB waived the requirement for informed consent for this retrospective study of medical records. 

JOURNAL REQUIREMENT 2: Thank you for stating the following in your Competing Interests section: “Drs. Erica Farrand, Michael Guarnieri, George Minowada, Mr. Lawrence Block, and Ms. Mei Lee have declared that no competing interests exist. Dr. Harold Collard has read the journal's policy and has the following competing interests: personal fees from Advanced Medical, Polarean and WebMD, and grants from Pulmonary Fibrosis Foundation outside of the submitted work.”

RESPONSE TO REQUIREMENT 2: The cover letter has been updated to include the Competing Interests statement in accordance with the online author guidance It now reads:

We have read the journal’s policy and one of the authors of this manuscript (Dr. Harold Collard) has the following competing interests: personal fees from Advanced Medical, Polerean, and WebMD, and grants from Pulmonary Fibrosis Foundation outside of the submitted work. These entities provided no funding and had no role in the design, preparation, or submission of this manuscript. The competing interest does not alter our adherence to PLOS ONE policies on sharing data and materials. 

We were unable to update the online submission form to reflect this change but appreciate you modifying it our behalf.

---

## [Decision Letter · Decision Letter 1]

1 Dec 2022

PONE-D-22-15208R1Extracting Patient-Level Data from the Electronic Health Record: Expanding Opportunities for Health System ResearchPLOS ONE

Dear Dr. Farrand,

Thank you for submitting your manuscript to PLOS ONE. After careful consideration, we feel that it has merit but does not fully meet PLOS ONE’s publication criteria as it currently stands. Therefore, we invite you to submit a revised version of the manuscript that addresses the points raised during the review process.

We look forward to receiving your revised manuscript.

Kind regards,

Nguyen Quoc Khanh Le

Academic Editor

PLOS ONE

Journal Requirements:

Reviewers' comments:

Reviewer's Responses to Questions

**Comments to the Author**

1. If the authors have adequately addressed your comments raised in a previous round of review and you feel that this manuscript is now acceptable for publication, you may indicate that here to bypass the “Comments to the Author” section, enter your conflict of interest statement in the “Confidential to Editor” section, and submit your "Accept" recommendation.

Reviewer #2: (No Response)

2. Is the manuscript technically sound, and do the data support the conclusions?

Reviewer #2: Yes

3. Has the statistical analysis been performed appropriately and rigorously? 

Reviewer #2: Yes

4. Have the authors made all data underlying the findings in their manuscript fully available?

Reviewer #2: Yes

5. Is the manuscript presented in an intelligible fashion and written in standard English?

Reviewer #2: Yes

6. Review Comments to the Author

Reviewer #2: Thank you for your revision. I see your comments addressing the editors' and journal's formatting and language around COI and ethics approval, but I don't see changes made regarding the reviewers' critiques and suggested changes from the first review.

7. PLOS authors have the option to publish the peer review history of their article (what does this mean?). If published, this will include your full peer review and any attached files.

Reviewer #2: No

---

## [Author Response · Author response to Decision Letter 1]

2 Dec 2022

JOURNAL REQUIREMENT 1: Please provide additional details regarding participant consent. In the ethics statement in the Methods and online submission information, please ensure that you have specified (1) whether consent was informed and (2) what type you obtained (for instance, written or verbal, and if verbal, how it was documented and witnessed). If your study included minors, state whether you obtained consent from parents or guardians. If the need for consent was waived by the ethics committee, please include this information. 

RESPONSE TO REQUIREMENT 1: Our full ethics statement was previously included in the online submission form but not in the methods section. The statement has been added to the methods section and now reads:

Institutional review boards at the University of California San Francisco (#14-15459), and the KPNC Division of Research (#CN-15-2126-H_05) approved the study protocol. The primary dataset was deidentified prior to access. A subset of patient records was identified for algorithm validation. The IRB waived the requirement for informed consent for this retrospective study of medical records. 

JOURNAL REQUIREMENT 2: Thank you for stating the following in your Competing Interests section: “Drs. Erica Farrand, Michael Guarnieri, George Minowada, Mr. Lawrence Block, and Ms. Mei Lee have declared that no competing interests exist. Dr. Harold Collard has read the journal's policy and has the following competing interests: personal fees from Advanced Medical, Polarean and WebMD, and grants from Pulmonary Fibrosis Foundation outside of the submitted work.”

RESPONSE TO REQUIREMENT 2: The cover letter has been updated to include the Competing Interests statement in accordance with the online author guidance It now reads:

We have read the journal’s policy and one of the authors of this manuscript (Dr. Harold Collard) has the following competing interests: personal fees from Advanced Medical, Polerean, and WebMD, and grants from Pulmonary Fibrosis Foundation outside of the submitted work. These entities provided no funding and had no role in the design, preparation, or submission of this manuscript. The competing interest does not alter our adherence to PLOS ONE policies on sharing data and materials. 

We were unable to update the online submission form to reflect this change but appreciate you modifying it our behalf. 

JOURNAL REQUIREMENT 3: Please ensure that your manuscript meets PLOS ONE's style requirements, including those for file naming. 

RESPONSE TO REQUIREMENT 3: The paper has been reformatted to meet PLOS ONE’s style requirements. 

JOURNAL REQUIREMENT 4: In your Data Availability statement, you have not specified where the minimal data set underlying the results described in your manuscript can be found. PLOS defines a study's minimal data set as the underlying data used to reach the conclusions drawn in the manuscript and any additional data required to replicate the reported study findings in their entirety. All PLOS journals require that the minimal data set be made fully available. For more information about our data policy, please see http://journals.plos.org/plosone/s/data-availability.

RESPONSE TO REQUIREMENT 4: The title page has been updated to include our Data Availability Statement with instructions on requesting access to the data set. 

“Data Availability: The dataset used in this study is held by the Kaiser Permanente Northern California Division of Research. Any researcher interested in accessing this dataset can submit an application form through the Research Collaboration Portal (rcp.kaiserpermanente.org) requesting access. Please contact the research collaboration portal staff (Email: Victoria.k.peckham@kp.org) for further assistance. All relevant data are within the paper.”

JOURNAL REQUIREMENT 5: Please include captions for your Supporting Information files at the end of your manuscript, and update any in-text citations to match accordingly. Please see our Supporting Information guidelines for more information: http://journals.plos.org/plosone/s/supporting-information.

RESPONSE TO REQUIREMENT 5: The supporting information captions and in-text citations now match the Supporting Information guidelines. 

JOURNAL REQUIREMENT 6: Please review your reference list to ensure that it is complete and correct. If you have cited papers that have been retracted, please include the rationale for doing so in the manuscript text or remove these references and replace them with relevant current references. Any changes to the reference list should be mentioned in the rebuttal letter that accompanies your revised manuscript. If you need to cite a retracted article, indicate the article’s retracted status in the References list and also include a citation and full reference for the retraction notice.

RESPONSE TO REQUIREMENT 6: We have added three additional references (see Response to Reviewer 2 Comment 1). The formatting for references has been updated to follow PLOS guidelines. 

Reviewer 1 Comments:

REVIEWER 1 COMMENT 1: The use of several data sources is a real strength, as is the wide range of data elements. The paper is well written. While they used NLP on the radiology reports, it was unclear to me if they used NLP on the EHR physician notes. Please clarify that. If not, why not? (This is not intended to be a barrier to publication, but we should know this.)

RESPONSE TO REVIEWER 1 COMMENT 1: 

We appreciate Reviewer 1’s recognition of the overall merits of the data set we have constructed. We share Reviewer 1’s interest in applying NLP broadly to unstructured data sources, and physician notes are a rich source of information. However, the complexity, length, and volume of physician notes requires development of more advanced NLP that integrates semantic and contextual understanding. We therefore elected to focus our initial application of NLP to radiology and pathology reports, which tend to be shorter in length and follow a more standardized format. Given the success with NLP application observed in this study, we plan to continue to expand our use of NLP in future studies. 

REVIEWER 1 COMMENT 2: The word data is a plural word. They write "the pharmacy data was queried..." No. it should be: "the pharmacy data were queried..."

RESPONSE TO REVIEWER 1 COMMENT 2: 

The referenced grammatical errors have been edited. 

Reviewer 2 Comments:

REVIEWER 2 COMMENT 1: First, has a similar demonstration ever been done with regional EHR for a different disease, even outside of pulmonary medicine (cancer, cardiology, or rheumatology, for example)? Your discussion should comment on this and put this paper in context with what has already been done (or not done) in this area.

RESPONSE TO REVIEWER 2 COMMENT 1: Regional EHR data has been very impactful in driving health services delivery research in other disease contexts. We appreciate Reviewer 1 raising this point and have made the following modification to the background section as well as included three references highlighting three recent impactful delivery science studies using data from the KPNC EHR. 

“Regional EHR data, with detailed patient-level information, has been particularly impactful in advancing delivery science in other contexts and stands to fundamentally improve population research in ILD(22-24).”

REVIEWER 2 COMMENT 2: There was a very low number of patients with “IPF” who were reported to take antifibrotic medications. Please discuss. Was this due to lack of access to (or use of) ILD specialty centers?

RESPONSE TO REVIEWER 2 COMMENT 2: The type of data we were able to extract from the EHR using automated methods enabled us to identify potential evidence-practice gaps in ILD care, of which the underutilization of antifibrotics was a significant one. Understanding the reason for this gap (and others) is outside of the scope of this study. However we are planning a follow up mixed-methods study to identify the patient, provider, and system-level barriers adopting evidence-based ILD practices in real-world settings. 

REVIEWER 2 COMMENT 3: The information included in “autoimmune serology” is limited. Guideline directed evaluation would include some additional tests (SSA, SSB, Jo-1 to name a few). What percentage had a full cadre of autoimmune serologies completed? What percent of the “IPF” patients had this evaluation?

RESPONSE TO REVIEWER 2 COMMENT 3: We agree with Reviewer 2 that our evaluation of autoimmune serologies was limited. To our knowledge, there are no international recommendations for serologic evaluation in a general ILD population, however ATS/ERS/JRS/ALAT have made recommendations for serologic studies in the diagnostic workup for idiopathic pulmonary fibrosis which we used to guide our data collection. Most of the patients in our cohort were diagnosed prior to 2018, we therefore relied on the official ATS/ERS/JRS/ALAT statement from 2011 which recommended testing for ANA, Rheumatoid Factor and Anti-CCP. In 2018, the recommendation was updated to include CRP, ESR, and a myositis panel, with “other detailed tests performed on a case-by-case bases according to associated symptoms and signs”. Since this updated recommendation was released during our study period and would only apply to a minority of patients, we decided to limit our evaluation to those studies specified in the 2011 statement. Furthermore, we found that given the broad clinical use of ESR and CRP, we could not discern whether their use was part of an ILD evaluation without further investigation (e.g. detailed chart review) which was outside of the scope of this study. Within the KPNC system, the myositis panel is often a send out lab and could not be reliably extracted. For all of these reasons we limited our serologic evaluation to ANA, RF and anti-CCP. We have added the following line to the methods section to provide the reader with this context:

“Autoimmune serologies were limited to antinuclear antibodies, rheumatoid factor, and anti-cyclic citrullinated peptide, three tests recommended as part of a general serologic evaluation in patients with suspected interstitial lung disease that could be reliably extracted from the EHR.”

We also now qualify the autoimmune serologies as “limited”.

REVIEWER 2 COMMENT 4: Are there any specialty ILD programs in the KPNC system in this study? If so, how often were they engaged and were there differences in any of your findings based upon whether or not they had at least one visit with an ILD program? If not, is there any way to know from your data or from other past publications how often KPNC patients are seen at a regional ILD center?

RESPONSE TO REVIEWER 2 COMMENT 4: KPNC does not have a specialty ILD program. There is an optional regional multidisciplinary conference. Following the identification of gaps in evidence-based practices, we are collaborating with our KPNC clinical colleagues to evaluate whether there are differences in diagnostic and management outcomes between ILD patients who undergo MDC review at KPNC versus those that do not. 

REVIEWER 2 COMMENT 5: On a similar note, it is quite surprising to me that 20% of patients with ILD did not see a pulmonologist (you state in the text that it was 4/5 that saw a pulmonologist – although Figure 2 shows that 100% of them saw a pulmonologist in year 1). Please clarify and discuss. Is there any clue that perhaps they saw a pulmonologist outside of the KPNC system? Was the significance of ILD not appreciated by the PCP or was the finding perhaps not even taken not of (I see this sometimes when a patient gets the CT for another reason such as a chest pain, nodule or aorta evaluation that the ILD never gets evaluated and the ordering physician just focuses on the initial indication for the scan). Could a sample of them have a chart review to learn if there was some explanation for this?

RESPONSE TO REVIEWER 2 COMMENT 5: The referenced statement in the discussion section summarizes health care utilization trends over the study period (8 years). On average, 80% of patients saw a pulmonologist annually. Figure 2 provides more granular data by breaking down health care utilization metrics by each year following diagnosis. Here we notice that 100% of patients saw a pulmonologist in the first year, and that number dropped to around 80% for subsequent years – consistent with the summary statement in the text. 

Reviewer 2 raises a good point that our evaluation of health care utilization is limited by our inability to robustly capture care accessed outside of the KPNC. We tried to mitigate this limitation by performing the study in an integrated community healthcare system which is less porous than a tertiary care center. However this remains a limitation of the study, which we now specifically name in the discussion section. 

“Third, limitations in our data set precluded analysis of other worthwhile topics including the types of physicians providing care (generalist vs specialist), the impact of multidisciplinary case conference discussion on the diagnosis and management of ILD, important utilization metrics including pulmonary rehabilitation and palliative care, and the extent to which patients accessed ILD care outside of the KPNC system.”

REVIEWER 2 COMMENT 6: What were the results of the subset of patients that were chart reviewed by an ILD specialist? I can't find this information, sorry if I am missing it. I see the information about the CT review but thought there was also intended to be a review of the clinical cases by chart review.

RESPONSE TO REVIEWER 2 COMMENT 6: 

A limited chart review was done to evaluate the performance of the ILD algorithm and NLP algorithm. To clarify this point to the reader we modified the following sentences in the methods section:

“A random validation sample of 200 cases underwent a structured medical record review by an expert ILD clinician (E.F.) to confirm ILD diagnosis and ILD subtype in order to assess performance of the ILD algorithm.” 

“A subset of NLP results (40%) was manually reviewed by an ILD expert (E.F.) and validated in order to assess the performance of the NLP algorithm.”

REVIEWER 2 COMMENT 7: You state that “the epidemiology, diagnostic evaluation, and management utilization mirrors many of the findings from tertiary expert center”. I challenge that statement some, given differences in the use of HRCT, numbers of autoimmune serology tests, and frequency of PFT or spirometry use. And you have no data about it, but I strongly doubt that there was frequent utilization of multidisciplinary discussion (MDD) groups, which is central to the diagnostic evaluation at a tertiary ILD center. Please speak to this in your discussion and soften your statement if you agree. Also, the management seems to differ greatly from what would be done at a tertiary expert center.

RESPONSE TO REVIEWER 2 COMMENT 7: 

We agree that this is a nuanced statement. Specifically we were referring to the demographic composition of the cohort, disease prevalence, the high and sustained rates of health care utilization, and reliance on CT chests and PFTs for diagnosis and monitoring. However as Reviewer 2 points out there are some important gaps between clinical guidelines and the real world practices we observed, as highlighted in the discussion section. We have modified the statement in question, replacing the word “many” with “a number of”. It now reads:

The epidemiology, diagnostic evaluation, and management utilization mirrors a number of the findings from tertiary expert centers3,4,28-30 

REVIEWER 2 COMMENT 8: The number of patients with IPF who did not have a UIP pattern on CT was quite high. This begs me to wonder whether a whole lot of these patients really had some other ILD but were declared to have IPF but were wrongly diagnosed. Please comment. A “baseline characteristics” table in the supplement with just “IPF” patients might shed some light on this, as many recent IPF studies have shown very consistent characteristics and so we can see how your study’s “IPF patients” are similar or different from other cohorts.

RESPONSE TO REVIEWER 2 COMMENT 8: We agree with Reviewer 2 that the distribution of ILD subtypes and breakdown of radiologic patterns raises questions about the diagnostic certainty. The goal of this paper was to develop automated approaches to ILD data extraction from the EHR and use that to describe a community based ILD cohort. In future studies we intend to refine algorithms to classify ILD subtypes (i.e. IPF) in order to better understand whether the evidence-practice gaps we have identified are due to challenges with ILD diagnosis, ILD management, or a combination of the two. 

REVIEWER 2 COMMENT 9: Was there a high correlation between the CT reports that were manually reviewed and determined to have UIP pattern and the 11% that would have been declared as UIP through NLP? If not fully correlated, then that may have some impact upon the operating characteristics you describe for UIP by NLP.

RESPONSE TO REVIEWER 2 COMMENT 9: Yes, the correlation between NLP detected UIP pattern, and UIP pattern on manual review is summarized by the PPV for detecting a UIP pattern using NLP = 94.29% (90.70, 96.48%).

REVIEWER 2 COMMENT 10: Was there a high correlation between the CT reports that were manually reviewed and determined to have UIP pattern and the 11% that would have been declared as UIP through NLP? If not fully correlated, then that may have some impact upon the operating characteristics you describe for UIP by NLP. (duplicate comment)

RESPONSE TO REVIEWER 2 COMMENT 10: See above

REVIEWER 2 COMMENT 11: I find in my own practice that many radiology reports from a chest CT do not comment one way or another about presence of UIP. Did you consider having a sample of some of those (the ones with no comment one way or another about UIP) independently reviewed?

RESPONSE TO REVIEWER 2 COMMENT 11: We had a senior chest radiologist consult on our approach to evaluating CT reports with NLP (please see acknowledgements section). We reviewed a wide sample of CT reports in collaboration with radiology to identify what terms or combination of terms best captured the impression of the radiologist. 

REVIEWER 2 COMMENT 12: Were the PFT values available as discrete values in the EHR extraction, or were they manually extracted? I know that with my own EPIC EHR system, PFT values are not usually available as discrete values/fields, although that may well be different in another hospital system. Were spirometry values (ones that were not part of a full PFT) also available, as such?

RESPONSE TO REVIEWER 2 COMMENT 12: This is such an important observation. The ability to extract PFTs from the EHR is generally quite challenging as the discrete values are most consistently captured in unstructured data sources – most notably physician notes or scanned reports. Reviewer 2’s experience mirrors our own working with PFT data at a tertiary care center. One of the advantages of working within an integrated care system like KPNC, is that the majority of patients perform their PFTs at a Kaiser Facility and the results are extractable from EPIC flowsheets. This also applies to spirometry values. 

As we think about scaling automated extraction of EHR data, we anticipate applying NLP to physician notes to gather this information from other health systems. 

REVIEWER 2 COMMENT 13: Were spirometry values (ones that were not part of a full PFT) also available, as such?

RESPONSE TO REVIEWER 2 COMMENT 13: See above

Reviewer 3 Comments:

REVIEWER 3 COMMENT 1: age 8, line 193: the Authors state that "11% ... were classified as UIP patterns (correlating with an IPF diagnosis)...". As UIP radiological pattern is not always unequivocally bound to IPF diagnosis, I would suggest omitting to refer to the clinical term of IPF.

RESPONSE TO REVIEWER 3 COMMENT 1: We agree with this comment and have modified the line as advised. 

REVIEWER 3 COMMENT 2: Page 9, lines 206-209: it would be interesting to learn the prescription pattern in UIP vs non-UIP.

RESPONSE TO REVIEWER 3 COMMENT 2: Our intention is that following the success of the simplified NLP algorithm applied to CT Chest reports in this study, we will continue to develop and refine NLP algorithms to extract additional information from both CT reports and other unstructured data sources (i.e. physician notes). 

REVIEWER 3 COMMENT 3: Table 1: surprisingly, rural geographic location was close to 0, and the Authors estimated the frequency of chronic HP to be 10% (the third most common diagnosis). This discrepancy may be worth commenting on.

RESPONSE TO REVIEWER 3 COMMENT 3: We agree with Reviewer 3 that this is an interesting finding. This study was not designed to comment on specific findings by ILD subtype or geographic location. However, in future studies we intend to refine algorithms to classify ILD subtypes in order to support more detailed epidemiologic studies. 

REVIEWER 3 COMMENT 4: Table 3: please consider ordering medications and monitoring tests according to frequency.

RESPONSE TO REVIEWER 3 COMMENT 4: We appreciate this recommendation and have modified Table 3 as advised.

---

## [Editor Report · Decision Letter 2]

27 Dec 2022

Extracting Patient-Level Data from the Electronic Health Record: Expanding Opportunities for Health System Research

PONE-D-22-15208R2

Dear Dr. Farrand,

We’re pleased to inform you that your manuscript has been judged scientifically suitable for publication and will be formally accepted for publication once it meets all outstanding technical requirements.

Kind regards,

Nguyen Quoc Khanh Le

Academic Editor

PLOS ONE
---

## [Editor Report · Acceptance letter]

9 Jan 2023

PONE-D-22-15208R2 

Extracting patient-level data from the electronic health record: expanding opportunities for health system research 

Dear Dr. Farrand:

I'm pleased to inform you that your manuscript has been deemed suitable for publication in PLOS ONE. Congratulations! Your manuscript is now with our production department. 

Kind regards, 

on behalf of

Dr. Nguyen Quoc Khanh Le 

Academic Editor

PLOS ONE